# LUNA: LANGUAGE AS CONTINUING ANCHORS FOR REFERRING EXPRESSION COMPREHENSION

## ABSTRACT

Referring expression comprehension aims to localize the description of a natural language expression in an image. Using location priors to remedy inaccuracies in cross-modal alignments is the state of the art for CNN-based methods tackling this problem. Recent Transformer-based models cast aside this idea making the case for steering away from hand-designed components. In this work, we propose LUNA, which uses language as continuing anchors to guide box prediction in a Transformer decoder, and show that language-guided location priors can be effectively exploited in a Transformer-based architecture. Specifically, we first initialize an anchor box from the input expression via a small "proto-decoder", and then use this anchor as location prior in a modified Transformer decoder for predicting the bounding box. Iterating through each decoder layer, the anchor box is first used as a query for pooling multi-modal context, and then updated based on pooled context. This approach allows the decoder to focus selectively on one part of the scene at a time, which reduces noise in multi-modal context and leads to more accurate box predictions. Our method outperforms existing state-of-the-art methods on the challenging datasets of ReferIt Game, RefCOCO/+/g, and Flickr30K Entities.

## 1 INTRODUCTION

Referring expression comprehension (REC) is the task of localizing natural language in images, where an expression in plain text describes a single or a group of objects from an image, and the objective is to put a bounding box around the target. It provides fundamental values to real-world applications such as robotics Tellex et al. (2020), image editing Shi et al. (2021), and surveillance Li et al. (2017). With purposes of reducing the search space and mitigating difficulties in cross-modal context modeling, early methods rely on prior knowledge about the "likely" locations of the target.

Specifically, region proposals and anchor boxes are the two most common types of location priors. Two-stage models Hu et al. (2017); Wang et al. (2018); Yu et al. (2018a;b); Yang et al. (2019a) leverage region proposals, consisting of thousands of massively overlapping boxes extracted using a standalone proposal method Uijlings et al. (2013); Zitnick & Dollár (2014); Ren et al. (2015). A cross-modal similarity-based ranking method is used to select one proposal as the prediction. Such models cannot recover from proposal failures and generally suffer a low recall rate Yang et al. (2019b). On the other hand, one-stage models Yang et al. (2019b; 2020); Luo et al. (2020) leverage dense anchor boxes defined over image locations and directly predict a bounding box from integrated multi-modal feature maps. This approach allows box regression to be conditioned on the input expression, thus addressing the aforementioned issue of two-stage models. The definitions of anchors however, are heuristic and greatly influence model performance.

The most recent Transformer-based methods Deng et al. (2021); Li & Sigal (2021) discard location priors and leverage the powerful correlation modeling power of the Transformer architecture Devlin et al. (2019); Vaswani et al. (2017) for end-to-end prediction. Typically, a Transformer encoder Devlin et al. (2019) jointly embeds visual and linguistic inputs, and box prediction is made from a context feature vector globally pooled from the encoder outputs based on query-to-feature similarities. The query (which can be a black-box learnable feature vector Deng et al. (2021) or a linguistic feature vector summarizing the expression Li & Sigal (2021)) serves as a representation of the target and context pooling is guided by similarity-based attention.

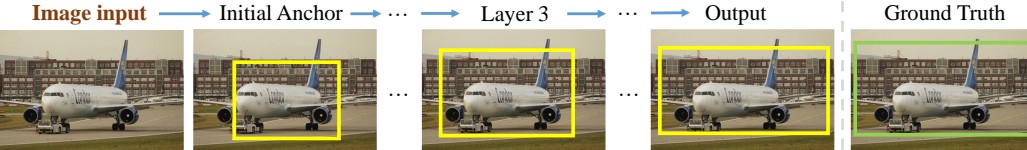

Figure 1: Example anchor boxes learned by our method. The anchor boxes are generated sequentially by a stack of modified Transformer decoder layers. The next anchor box is predicted based on context aggregated by the current anchor box. The last predicted box is used as the final prediction.

A potential issue with this approach is that without location priors (pre-designed or learnable), these queries rely on pure "content"-based similarities to decide context locations, which often contain numerous inaccuracies in practice. Li *et al.* Li & Sigal (2021) demonstrate that even in scenes of relatively simple compositions, the attention of the context feature vector from this approach often peaks at multiple locations, including those outside the target area, which leads to wrong or inaccurate box predictions.

Motivated by the above observations, in this work we propose a Transformer-based decoding method for addressing REC, which we term as LUNA (short for LangUage as contiNuing Anchors). It consists in leveraging the input expression for generating a series of continuously updated anchor boxes (shown in Fig. 1) that guide object localization. LUNA generates the first anchor box by attending to image regions under the guidance of the input expression. This is achieved via a cross-attention-based proto-decoder, which summarizes an object representation based on word-specific visual context and decodes an approximate location of the target. Given the initial object representation and the anchor box, a stack of modified Transformer decoder layers iteratively refine the object representation and update the anchor box. Progressing through each layer, the current anchor box is projected into high-dimensional space and used for pooling multi-modal context; then a new anchor box is predicted from pooled context. We refer to this stack of decoder layers as a continuous anchor-guided decoder. This decoding approach allows the model to focus selectively on one part of the scene at a time, and acquire more accurate context information with focused attention.

To evaluate the efficacy of the proposed method, we conduct extensive experiments on the challenging datasets of ReferIt Kazemzadeh et al. (2014), RefCOCO Yu et al. (2016), RefCOCO+ Yu et al. (2016), RefCOCOg Nagaraja et al. (2016), and Flickr30K Entities Plummer et al. (2015), for which we improve the state of the art by large margins.

Our contributions are summarized as follows. (1) We proposed a simple but effective Transformer-based decoding method which leverages anchors as position priors for tackling referring expression comprehension. (2) Our decoding strategy consists of a proto-decoder to estimate the initial localization from language, and a continuous anchor-guided decoder for predicting bounding boxes progressively. (3) We obtain new state-of-the-art results on five REC benchmarks, demonstrating the effectiveness and generality of the proposed method.

## 2 RELATED WORK

**Referring expression comprehension (REC).** Since the proposal of this task in several parallel studies Yu et al. (2016); Hu et al. (2016); Nagaraja et al. (2016), the state-of-the-art paradigm for tackling it has transitioned from *two-stage* models to *one-stage* models to the most recent *Transformer-based* models. Two-stage models Hu et al. (2017); Wang et al. (2018); Yu et al. (2018a;b); Yang et al. (2019a) rank a set of image regions based on their similarities with the referring language expression, where the regions are pre-extracted using a region proposal method. Popular proposal methods include unsupervised ones based on hand-crafted features Uijlings et al. (2013); Zitnick & Dollár (2014) and pre-trained neural nets Ren et al. (2015). This propose-and-rank approach is slow and suffers from limitations of the proposal method (such as location inaccuracies from unsupervised methods or biases of the proposal network) Liao et al. (2020); Yang et al. (2019b).

To address these issues, one-stage models opt for directly predicting the bounding box from fused multi-modal features, relying on feature fusion mechanisms and dense anchoring Yang et al. (2019b;

2020); Luo et al. (2020). Typically, a cross-modal feature fusion module (such as concatenation-based Yang et al. (2019b; 2020) or attention-based Luo et al. (2020)) integrates extracted visual and linguistic features to generate multi-modal features. Then a set of reference boxes, termed anchor boxes, are densely placed at different locations of the multi-modal feature maps, and boxes are predicted with respect to anchors in a sliding window fashion. By directly conditioning box predictions on the referring expression, one-stage models circumvent the problems faced by their two-stage precursors. However, the heuristics used for defining the anchor boxes (concerning their scales, aspect ratios, and assignments) can significantly influence the accuracy of the model. Another line of work (*e.g.*, Liao et al. (2020)) leverages language as correlation filters and does not resort to anchors, but has not reached the accuracy of anchor-based methods.

Most recently, the powerful Transformer Devlin et al. (2019); Vaswani et al. (2017) architecture is adapted to this problem. The core ingredients of Transformer-based methods Deng et al. (2021); Li & Sigal (2021) include a Transformer encoder for jointly embedding visual and linguistic features, and a cleverly designed "query", which aggregates multi-modal context used for box prediction. In a ViT-like Dosovitskiy et al. (2021) manner, Deng *et al.* Deng et al. (2021) leverage a special [REG] feature vector prepended to the input sequence as the query, which pools global context in the encoder's self-attention layers. In a DETR-like Carion et al. (2020) manner, Li *et al.* Li & Sigal (2021) employ a Transformer decoder, and use a linguistic feature vector (which summarizes the input expression) as a query in the decoder. Either the [REG] token approach or the language-as-query approach relies on purely content-based similarities for context aggregation, which tend to have many inaccuracies due to noises in the image and expression. Motivated by previous one-stage models and recent Transformer-based detection models that exploit spatial representations for decoding Zhu et al. (2021); Liu et al. (2022), in this work we leverage learnable anchor boxes as queries for helping guide multi-modal context aggregation. Our model learns more accurate cross-modal alignments compared to models in Deng et al. (2021) and Li & Sigal (2021).

**Multi-modal understanding**. One major line of research in the area of multi-modal (vision-language) understanding is the large-scale pre-training of multi-modal representations transferable to a variety of downstream tasks. This problem is typically tackled by leveraging large amounts of paired image-text data (typically sourced from the Internet) and training a Transformer-based model for solving various surrogate tasks, including masked cross-modal modeling Lu et al. (2019); Tan & Bansal (2019); Zhou et al. (2020), contrastive learning Radford et al. (2021), and modulated detection Kamath et al. (2021). Another line of work Lu et al. (2020); Hu & Singh (2021) takes a multi-task learning perspective and aims to develop a unified model that can address a large number of vision-language tasks at once. In addition, growing efforts have been devoted to a variety of vision-language tasks such as visual question answering Antol et al. (2015), image captioning Vinyals et al. (2015), and temporal sentence grounding Anne Hendricks et al. (2017); Gao et al. (2017). In this work, we focus on the task of referring expression comprehension and propose an anchor-based decoding mechanism based on the Transformer architecture.

## 3 METHOD

We are interested in the problem of box prediction from language, where the object of interest is represented in two different modalities. Therefore, our model should perform well in two aspects: aligning the representation of the object across the two modalities, and determining its precise location on the image. We first exploit the global correlation modeling power of a Transformer encoder Devlin et al. (2019) for learning fully connected pairwise correspondences between each image patch and each word (please see Sec.3.1). To localize the language precisely, we leverage a series of iteratively updated anchor boxes as location priors for guiding multi-modal context aggregation, which is achieved with a proto-decoder and a continuous anchor-guided decoder described in Sec. 3.2. We name the "proto-decoder" this way to give relevance to its functionality to output an initial representation of the object and an initial estimate of the object location. The overall pipeline of our model is schematically illustrated in Fig. 2.

### 3.1 VISUAL-LINGUISTIC ENCODING

We extract visual features, $\mathcal{F}_v \in \mathbb{R}^{HW \times C}$, from the input image by leveraging a vision backbone network (such as ResNet101 He et al. (2016) or Swin Transformer Liu et al. (2021)) followed by a

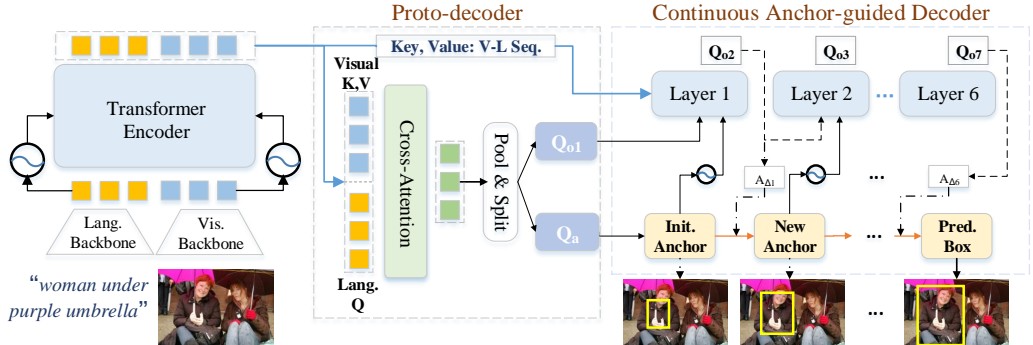

Figure 2: Overall pipeline of the proposed method. Our model consists of three main components: (1) a visual-linguistic Transformer encoder (Sec. 3.1) which jointly encodes the image and the text inputs, (2) a proto-decoder (Sec. 3.2) which produces an initial representation of the object and an initial estimate of its location (the first anchor box), and (3) a continuous anchor-guided decoder (also Sec. 3.2) which sequentially updates the anchor box through a stack of modified Transformer decoder layers. At each layer, both the output embedding from the previous layer and the anchor box are used as queries. The last anchor box corresponds to the final prediction.

multi-layer perceptron (MLP) for channel reduction. Here, $C$ denotes the channel number, and $H$ and $W$ denote the height and width, respectively. And we extract linguistic features, $\mathcal{F}_l \in \mathbb{R}^{T \times C}$, from the input expression by leveraging a language modeling network (such as LSTM Hochreiter & Schmidhuber (1997) or BERT Devlin et al. (2019)) followed by an MLP for channel reduction. Here, $T$ denotes the number of words and $C$ denotes the number of channels. The visual features $\mathcal{F}_v$ and linguistic features $\mathcal{F}_l$ are concatenated along the sequence dimension to form the input to our Transformer encoder, $\mathcal{S} = [\mathcal{F}_v; \mathcal{F}_l]$, which is a visual-linguistic sequence of feature vectors. In the rest of the paper, we use "feature vector" and "embedding" interchangeably.

**The multi-modal Transformer encoder.** A multi-modal Transformer encoder takes $S$ as input, and outputs a sequence of multi-modal embeddings which align visual cues and linguistic meanings in a common feature space. We denote the encoder output as $\mathcal{E} \in \mathbb{R}^{(HW+T) \times C}$, where $H$, $W$, $T$, and $C$ follow their previous definitions. Our encoder mostly follows the original implementation of the BERT$_{\text{BASE}}$ encoder Devlin et al. (2019), which consists of a stack of 6 identical layers. Each layer mainly includes a multi-head self-attention sub-layer and a feed-forward sub-layer. To encode position information and also differentiate between the two types of inputs, we use positional embeddings, denoted as $\mathcal{P} = [\mathcal{P}_v; \mathcal{P}_l]$, where $\mathcal{P}_v \in \mathbb{R}^{HW \times C}$ are fixed image positional embeddings the same as in Carion et al. (2020) and $\mathcal{P}_l \in \mathbb{R}^{T \times C}$ are learnable expression positional embeddings Li & Sigal (2021). $\mathcal{P}$ is supplemented to each attention sub-layer.

## 3.2 Decoding with continuing anchors

As described in Secs. 1 and 2, previous Transformer-based methods Deng et al. (2021); Li & Sigal (2021) tackling this task do not exploit location priors during decoding. As illustrated in Fig. 4, this often leads to noise in the aggregated context feature vector. We use a series of "continuing" anchors (in the sense that each anchor box is updated from the previous one) as queries during decoding for addressing this problem.

**Query generation with a proto-decoder.** The proto-decoder (illustrated in Fig. 3 (a)) generates the first anchor box and an object representation (what we refer to as the object query) from multi-modal context. Without introducing extra human-designed strategy, our learnable anchor could naturally perform a role of location prior.

To start with, word-specific multi-modal context is computed via a cross-attention layer. The query input to this layer is a concatenation of $\mathcal{E}_l \in \mathbb{R}^{T \times C}$ and $\mathcal{P}_l$ (the expression positional embeddings described in Sec. 3.1) along the channel dimension. Here, $\mathcal{E}_l$ is a segment of the encoder output $\mathcal{E}$ from positions corresponding to word embeddings. The key and the value inputs to this layer is a concatenation of $\mathcal{E}_v \in \mathbb{R}^{HW \times C}$ and $\mathcal{P}_v$ (the image positional embeddings described in Sec. 3.1) along the channel dimension. Similarly, $\mathcal{E}_v$ is a segment of $\mathcal{E}$ from positions corresponding to image

region embeddings. The output from the cross-attention layer encodes word-specific multi-modal context and is denoted as $O \in \mathbb{R}^{T \times 2C}$. Next, average pooling is applied on $O$ across the sequence dimension to obtain the pooling output $Q \in \mathbb{R}^{1 \times 2C}$.

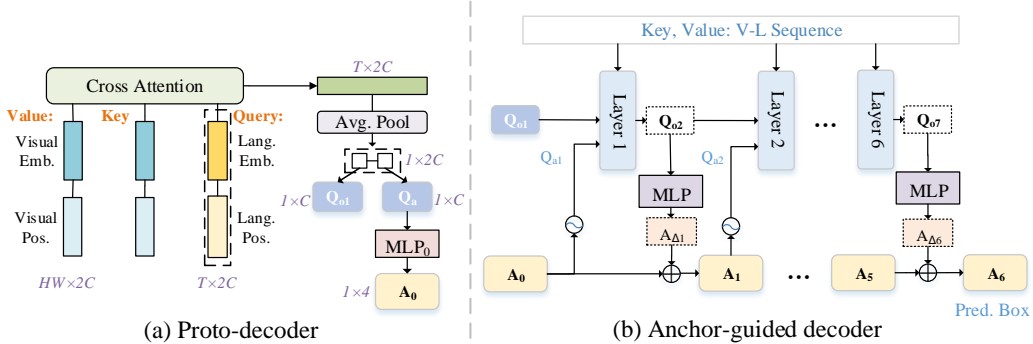

Figure 3: Details of the proto-decoder and the continuous anchor-guided decoder. (a) Proto-decoder generates the first learnable anchor query and object query. (b) Anchor-guided decoder refine the anchor query after multi-modal context aggregation of each layer.

In a final step, we split $Q$ along the channel dimension to obtain two feature vectors: $Q_{o1} \in \mathbb{R}^{1 \times C}$ and $Q_a \in \mathbb{R}^{1 \times C}$. $Q_{o1}$ is the object query we want. And $Q_a$ is projected by an MLP with 4 output channels and a terminal sigmoid activation. The output is our first anchor box, denoted as $A_0 \in \mathbb{R}^{1 \times 4}$. The 4 values along the channel dimension have a physical meaning and can be represented as a 4-tuple: $(x_0, y_0, w_0, h_0)$, where $x_0$ and $y_0$ are the coordinates of the center point, and $w_0$ and $h_0$ are the height and width of the anchor, respectively.

**Continuous anchor-guided decoding.** To provide better guidance for referring localization, inspired by the DAB-DETR Liu et al. (2022) detection model, we propose a continuous anchor-guided decoder, which uses the anchor query $A_0$ as an initial bounding box and iteratively updates it based on multi-modal context aggregation, as illustrated in Fig. 3 (b).

Assuming there are $n$ decoder layers (original transformer decoder layer which contains a self-attention and a cross-attention sub-layers), we use $i \in [1, 2, ...n]$ to index each layer. Given an anchor box $A_{i-1}$, we project it to high-dimensional space to obtain an anchor query as follows,

$$Q_{ai} = \theta([\text{PE}(x_{i-1}); \text{PE}(y_{i-1}); \text{PE}(w_{i-1}); \text{PE}(h_{i-1})]), \tag{1}$$

where '[;]' indicates concatenation along the channel dimension, PE is a sinusoidal positional encoding function as we describe next, and $\theta$ is a fully connected layer with $C$ output channels. To put it into words, to convert the anchor box into a high-dimensional embedding for use as a query, we first apply a positional encoding function which operates element-wise on each of the 4 parameters of the box and embeds each parameter into a high-dimensional embedding; then we concatenate the four embeddings along the channel dimension and projects it. Note that $\theta$ is shared across all layers. We adopt the same positional encoding function as defined in Devlin et al. (2019):

$$\text{PE}(z)_{2j} = \sin\left(\frac{z}{\mathcal{T}^{2j/D}}\right), \quad \text{PE}(z)_{2j+1} = \cos\left(\frac{z}{\mathcal{T}^{2j/D}}\right), \tag{2}$$

where $D$ defines the number of output channels, $j$ is the index along the channel dimension, $z$ is the input position variable, and $\mathcal{T}$ represents temperature. We discuss the hyper-parameters $D$ and $\mathcal{T}$ in Sec. 4.2. Note that our proposed anchor formulation is different from DAB-DETR, where we project 4 parameters (including height and width) of anchors while DAB-DETR embeds first 2 parameters (coordinates of the center point). We demonstrate the experimental effectiveness of our anchor query formulation in Supplementary Material due to the page limit.

The query input to the $i$-th cross-attention sub-layer is the following:

$$Q_i = [Q_{oi}; Q_{ai}], \tag{3}$$

where '[;]' denotes concatenation along the channel dimension, $Q_{oi} \in \mathbb{R}^{1 \times C}$ is the object query, and $Q_{ai} \in \mathbb{R}^{1 \times C}$ is the anchor query. With the multi-head attention mechanism in the cross-attention

sub-layer, the concatenation of these two types of queries decouples context aggregation Meng et al. (2021), *i.e.*, the output embedding, $Q_{o(i+1)} \in \mathbb{R}^{1 \times C}$, has the top/bottom half of channels generated based on the attention of the object query/anchor query, correspondingly. $Q_{o(i+1)}$ is the object query in the next decoder layer. To generate the next anchor box, we predict a set of offsets:

$$A_{\Delta i} = \sigma(\text{MLP}(Q_{o(i+1)})), \tag{4}$$

where $\sigma$ denotes the sigmoid function and $A_{\Delta i}$ is the set of box offsets. Note that the MLP is shared across all layers. We obtain a new anchor box by updating the previous one with the predicted offsets:

$$A_i = A_{i-1} + A_{\Delta i}, \tag{5}$$

We iterate through all $n$ layers of the decoder according to Eqs. equation 1-equation 5. The last anchor box, $A_n$, is used as the final prediction.

**The loss function.** We use the sum of the Generalized IoU Rezatofighi et al. (2019) loss and the L1 loss for training our model, which is defined as $\mathcal{L} = \lambda_0 \mathcal{L}_{\text{giou}}(B, \tilde{B}) + \lambda_1 \|B - \tilde{B}\|_1$, where $B$ is the ground truth, and $\tilde{B}$ is our prediction, $\lambda_0$ and $\lambda_1$ are balancing weights.

# 4 EXPERIMENTS

## 4.1 DATASETS AND METRICS

**RefCOCO, RefCOCO+, and RefCOCOg.** The RefCOCO dataset Yu et al. (2016) contains 19,994 images, 50,000 objects, and 142,209 referring expressions. The RefCOCO+ dataset Yu et al. (2016) contains 141,564 expressions for 49,856 objects in 19,992 images. The RefCOCOg dataset Nagaraja et al. (2016) contains 85,474 expressions for 54,822 objects in 26,711 images. Referring expression annotations in RefCOCO and RefCOCO+ are collected in an interactive two-player game. In this game, each player is motivated to provide a minimally sufficient description for the other to identify the target. As a result, expressions in these two datasets tend to be succinct, averaging fewer than 4 words per sentence. A special property of RefCOCO+ is that location words (such as "top", "bottom", "left", *etc.*) are banned in its annotations, which makes it the harder of the two. On the contrary, expressions in RefCOCOg are annotated on Amazon Mechanical Turk in a non-competitive way, and thus tend to be longer (8.43 words per sentence on average) and more expressive. This makes RefCOCOg a particularly challenging dataset compared to the other two.

**ReferIt Game.** The ReferIt Game dataset Kazemzadeh et al. (2014) contains 20,000 images collected from the SAIAPR-12 dataset Escalante et al. (2010) with automatically generated referring expressions. Following standard practice Hu et al. (2016); Yang et al. (2019b); Deng et al. (2021); Yang et al. (2020), we split this dataset into 9,000 images for training, 1,000 images for validation, and 10,000 images for testing, with 54K, 5.8K, and 60K referring expressions for these splits, respectively.

**Flickr30K Entities.** The Flickr30K Entities dataset Plummer et al. (2015) is a popular benchmark for the phrase grounding task, in which the model is required to localize multiple entities from an image caption. In our experiments, we follow a variant of the above setting which is commonly adopted by many previous REC methods Rohrbach et al. (2016); Yu et al. (2018b); Yang et al. (2019b; 2020); Yu et al. (2018a); Deng et al. (2021), *i.e.*, each phrase in a given image caption is treated as an independent referring expression without considering the context of the whole sentence. This dataset contains 31,783 images and 427K entities, with 29,783 images for training, 1,000 images for validation, and 1,000 images for testing.

**Evaluation metrics.** We adopt the standard metric of Acc@0.5, which measures the percentage of test samples for which the intersection-over-union between the prediction and the ground truth is above 0.5. This metric is also often referred to as top-1 accuracy or simply accuracy in the literature.

## 4.2 IMPLEMENTATION DETAILS

We use ResNet-101 He et al. (2016) and Swin-B Liu et al. (2021) as our visual backbone with classification weights pre-trained on ImageNet-22K. We adopt the uncased BERT$_{\text{BASE}}$ Devlin et al. (2019) model from the HuggingFace library Liu et al. (2019a) as our language model, and initialize

Table 1: Comparison with state-of-the-art methods on RefCOCO family datasets in terms of Acc@0.5. We indicate the image resolution for one-stage approaches.

| Method | Image Backbone | Image Res. | RefCOCO | | | RefCOCO+ | | | RefCOCOg | |
|---|---|---|---|---|---|---|---|---|---|---|
| | | | val | test A | test B | val | test A | test B | val | test |
| RvG-Tree Hong et al. (2019) | FRCN | - | 75.06 | 78.61 | 69.85 | 63.51 | 67.45 | 56.66 | 66.95 | 66.51 |
| MAttNet Yu et al. (2018a) | FRCN | - | 76.65 | 81.14 | 69.99 | 65.33 | 71.62 | 56.02 | 66.58 | 67.27 |
| DGA Yang et al. (2019a) | FRCN | - | - | 78.42 | 65.53 | - | 69.07 | 51.99 | - | 63.28 |
| NMTTree Liu et al. (2019a) | FRCN | - | 76.41 | 81.21 | 70.09 | 66.46 | 72.02 | 57.52 | 65.87 | 66.44 |
| CM-Att-Erase Liu et al. (2019b) | FRCN | - | 78.35 | 83.14 | 71.32 | 68.09 | 73.65 | 58.03 | 67.99 | 68.67 |
| DDPN Yu et al. (2018b) | FRCN | - | 76.8 | 80.1 | 72.4 | 64.8 | 70.5 | 54.1 | - | - |
| FAOA Yang et al. (2019b) | Darknet-53 | 256 | 72.54 | 74.35 | 68.50 | 56.81 | 60.23 | 49.60 | 61.33 | 60.36 |
| ReSC-Large Yang et al. (2020) | Darknet-53 | 256 | 77.63 | 80.45 | 72.30 | 63.59 | 68.36 | 56.81 | 67.30 | 67.20 |
| MCN Luo et al. (2020) | Darknet-53 | 416 | 80.08 | 82.29 | 74.98 | 67.16 | 72.86 | 57.31 | 66.46 | 66.01 |
| RCCF Liao et al. (2020) | DLA-34 | 512 | - | 81.06 | 71.85 | - | 70.35 | 56.32 | - | 65.73 |
| TransVG Deng et al. (2021) | ResNet-101 | 640 | 81.02 | 82.72 | 78.35 | 64.82 | 70.70 | 56.94 | 68.67 | 67.73 |
| RefTR Li & Sigal (2021) | ResNet-101 | 640 | 82.23 | 85.59 | 76.57 | 71.58 | 75.96 | 62.16 | 69.41 | 69.40 |
| SeqTR Zhu et al. (2022) | Darknet-53 | 640 | 81.23 | 85.00 | 76.08 | 68.82 | 75.37 | 58.78 | 71.35 | 71.58 |
| LUNA (Ours) | ResNet-101 | 640 | 84.47 | 86.64 | 80.18 | 72.60 | 77.90 | 64.51 | 74.06 | 72.75 |
| LUNA (Ours) | Swin-B | 640 | **86.12** | **88.43** | **82.63** | **75.96** | **80.62** | **68.36** | **76.51** | **76.55** |

it with official weights pre-trained on BooksCorpus Zhu et al. (2015) and English Wikipedia. All other parts of our model are randomly initialized. We adopt AdamW Loshchilov & Hutter (2019) as the optimizer, and set the learning rate to 0.0001 for the visual-linguistic Transformer encoder and 0.00005 for the visual and linguistic backbones. Following Deng et al. (2021) Li & Sigal (2021), images are resized to $640\times640$ and augmented with random intensity saturation and affine transformations. On RefCOCO, RefCOCO+, RefCOCOg, and ReferIt Game, the model is trained for 90 epochs with the learning rate dropped by a factor of 10 after 60 epochs. On Flickr30K Entities, the model is trained for 60 epochs with the learning rate dropped by a factor of 10 after 40 epochs. When adopting pre-training, we follow Li & Sigal (2021) and train our model on Visual Genome Krishna et al. (2017) for 6 epochs, and fine-tune it on the evaluation dataset for 50 epochs. The maximum expression length is 20 on all datasets. The visual-linguistic Transformer encoder and our decoder both have 6 layers. $C$, $\mathcal{T}$, and $D$ in Sec. 3 are empirically set to 256, 20, and 128, respectively. Models adopting a Swin-B backbone are trained with mini-batches of size 8, and those adopting a ResNet-101 backbone are trained with batch size 40. We apply balancing weights on loss function as $\lambda_0 = 2$, $\lambda_1 = 5$. We run several experiments with different random seeds, and the difference between different runs generally lies in $\pm0.5\%$.

## 4.3 COMPARISON WITH OTHERS

In Table 1, we compare LUNA to state-of-the-art methods on 3 standard benchmarks for the REC task. Among the existing method on REC, our method attains the highest Acc@0.5 across all subsets of all datasets with both ResNet-101 and Swin-B visual backbone. Specifically, on the val/test A/test B subsets of RefCOCO and RefCOCO+, LUNA with Swin-B backbone outperforms the second best method Li & Sigal (2021)by absolute margins of 3.89%/2.84%/6.06% and 4.38%/4.66%/6.20%, respectively. As for the ResNet-101 backbone, LUNA still achieves superior results by absolute margins up to 4.1%. On the most challenging RefCOCOg dataset, LUNA is able to achieve the largest improvements by far: On the validation set and test set, LUNA obtains 7.10% and 7.15% absolute improvements over the RefTR model, 5.16% and 4.97% absolute margin over second-best method SeqTR.

Observing the pattern across the three datasets, we can see that the more difficult the dataset is (please see Sec. 4.1 for a discussion), the larger the improvement LUNA is able to bring with respect to the state of the art. This corroborates our case for exploiting learnable location priors to facilitate box decoding. As discussed in Sec. 4.1, banning the use of location words makes Ref-COCO+ a significantly harder dataset than RefCOCO is, as cross-modal alignment can only rely on appearance information. Similarly, the longer and more flowery descriptions in the RefCOCOg dataset makes purely content-based similarities less reliable as the basis of grounding. The relatively large improvements of LUNA on these two datasets provide evidence that when content-based map-

ping is not easy or accurate, our proposed learnable location priors can make up for it. Moreover, on the ReferIt Kazemzadeh et al. (2014) dataset, LUNA also surpasses all previous methods.

In Table 2, we evaluate LUNA against the state-of-the-art methods on the Flickr30K Entities Plummer et al. (2015) dataset. Please refer to Sec. 4.1 for a discussion on the specific task setting. Compared to TransVG and RefTR, two other state-of-the-art models based on Transformers, our method obtains 2.04% and 2.48% (absolute) improvements in terms of Acc@0.5, respectively.

While pre-training is not the focus of this work, additional pre-training of LUNA on the Visual Genome Krishna et al. (2017) re-gion description splits can further boost performance, and our method obtains comparable or better results with respect to other methods that employ pre-training. More result and comparison please see Table 7 in Supplementary Material.

Table 2: Results on Flickr30K Entities and ReferIt.

| Method | Visual Backbone | Flickr30K test | ReferIt test |
|---|---|---|---|
| CITE Plummer et al. (2018) | VGG-16 | 61.89 | - |
| Similarity Net Wang et al. (2018) | VGG-16 | 60.89 | - |
| DDPN Yu et al. (2018b) | ResNet-101 | 73.30 | 63.00 |
| FAOA Yang et al. (2019b) | Darknet-53 | 68.69 | 59.30 |
| ReSC-Large Yang et al. (2020) | Darknet-53 | 69.28 | 64.60 |
| TransVG Deng et al. (2021) | ResNet-101 | 79.10 | 70.73 |
| RefTR Li & Sigal (2021) | ResNet-101 | 78.66 | 71.42 |
| LUNA (ours) | ResNet-101 | 79.44 | 72.67 |
| LUNA (ours) | Swin-B | **81.14** | **73.69** |

## 4.4 ABLATION STUDY

In this section, we conduct extensive ablation experiments to study the effects of the main model components and alternative design choices in the proto-decoder and the continuous anchor-guided decoder. Experiments in this section are conducted on the validation and test sets of the RefCOCOg dataset. ResNet-101 is adopted as the vision backbone and BERT is adopted as the language back-bone, unless otherwise specified.

Table 3: Model component ablations.

| Proto-decoder | CA-guided decoder | val | test |
|---|---|---|---|
| ✓ | ✓ | **74.06** | **72.75** |
| ✓ | | 69.58 | 69.82 |
| | ✓ | 71.26 | 70.55 |

Table 4: Ablations on proto-decoder.

| Methods | val | test |
|---|---|---|
| Self-attention | 70.42 | 70.36 |
| Reversed proto-decoder | 72.18 | 71.56 |
| Ours Proto-decoder | **74.06** | **72.75** |

**Model components.** In Table 3, we investigate the contributions of the proto-decoder and the contin-uous anchor-guided decoder (CA-guided decoder). We first remove the CA-guided decoder from the full model, and use the initial anchor produced from the proto-decoder as the prediction. This model variant leads to 4.48% and 2.93% drop in Acc@0.5 on the validation and test sets of RefCOCOg, respectively. These results validate the effectiveness of our proposed CA-guided decoder. We then remove the proto-decoder from the full model, and randomly initialized learnable embeddings of object query and anchor box which send to the CA-guided decoder. This model variant leads to 2.80% and 2.2% drop in Acc@0.5 on the validation and test sets of RefCOCOg, respectively. This demonstrates the effectiveness of the proposed proto-decoder.

**Alternative studies of proto-decoder.** We carefully designed the structure of proto-decoder to generate relyable initial object query and anchor query, and consider several replaceable cases (in-cluding self-attention, visual query *etc.*) as shown in Table 4. The result shows that our design of proto-decoder out-performs other alternative cases. Due to the page limit, more details please see Table 8 in Supplementary Material.

**Object representation in the proto-decoder.** As described in Sec. 3.2, after obtaining a sequence of word-specific visual context embeddings, we obtain an object representation by averaging the sequence for predicting the initial object query and anchor box. In Table 5, we study three other alternatives for summarizing object information. In the '[CLS] Token' alternative, we use the em-bedding output from the special '[CLS]' token input as representation of the object. In the 'Max Pooling' alternative, we directly pool the maximum activation along each token dimension across the sequence. As for 'Weighted Avg.' alternative, we apply weighted average pooling on the se-quence length dimension to generate the object representation, where weights are learned by 2-layer MLP with softmax activation. As shown in Table 5, normal average pooling out-performs other alternatives and we adopt it as the default choice.

Table 5: Alternative object representations in the proto-decoder.

| Method | val | test |
|---|---|---|
| CLS Token | 72.36 | 71.09 |
| Max Pooling | 71.73 | 71.69 |
| Weighted Avg. | 72.03 | 71.66 |
| Average Pooling | **74.06** | **72.75** |

Table 6: Effects of the number of layers in CA-guided decoder.

| # of Layers | val | test |
|---|---|---|
| 0 | 69.58 | 69.82 |
| 2 | 70.44 | 70.43 |
| 4 | 72.32 | 71.64 |
| 6 | **74.06** | **72.75** |
| 8 | 72.44 | 71.81 |

**The number of layers in CA-guided decoder.** In Table 6, we study the effects of the number of decoding layers in our proposed CA-guided decoder, *i.e.* the number of steps that initial anchor updated to predict the final bounding box. We test model variants which adopt 2, 4, 6, and 8 layers. Accuracy improves as more layers are added until 6 layers have been used. Note that in 0-layer case, LUNA only uses proto-decoder to predict the result.

## 4.5 VISUALIZATION

In Fig. 4, we visualize the attention patterns and predictions of LUNA and RefTR Li & Sigal (2021) on some challenging examples. Notice that for both models, the prediction is made where query attention is focused. This is expected as the box is predicted from a context feature vector pooled according to query attention. In a cluttered scene depicted on the left, our model correctly localizes the muffins among many similar distractors such as sandwiches in the middle or on top. In the scene on the right, the wing of the airplane is depicted from a rare angle and not quite noticeable from its background. Our model also correctly localizes the target in this case. In comparison, the query attention of RefTR peaks at wrong locations and leads to wrong box predictions.

*"the muffins on the bottom of the photo"*   *"the wing of an airplane over a bus"*

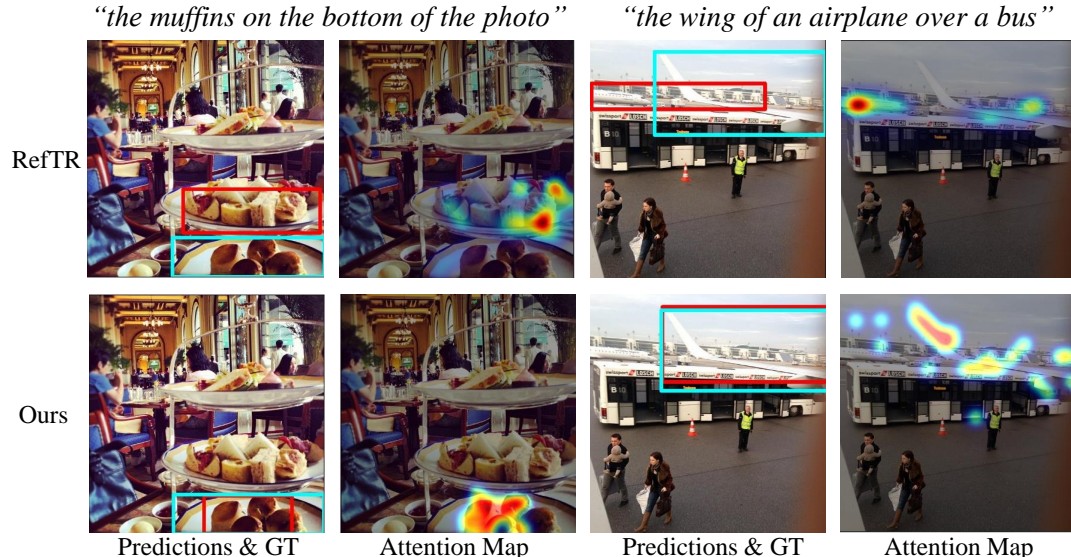

Figure 4: Visualized query attention from the last decoder layer and the final predictions of our method and RefTR Li & Sigal (2021). Cyan boxes are the ground truth and red boxes are predictions. Examples are from the validation set of RefCOCOg Nagaraja et al. (2016).

## 5 CONCLUSION

In this paper, we have presented a Transformer-based decoding method for referring expression comprehension, which exploits a series of language-guided anchor boxes as helpful spatial cues for guiding context pooling in a Transformer decoder. Extensive experiments on five benchmarks demonstrate its advantage with respect to the state-of-the-art methods.

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
