# OpenReview forum: "LUNA: Language as Continuing Anchors for Referring Expression Comprehension"
_ICLR.cc/2023/Conference — Submitted to ICLR 2023_

### Official Review · Reviewer_uU4Z · 2022-10-21

**Confidence:** 5
**Correctness:** 3
**Technical Novelty And Significance:** 3
**Empirical Novelty And Significance:** 3
**Recommendation:** 5

**Clarity, Quality, Novelty And Reproducibility:**

+ Clarity: The whole paper is well-organized and easy-to-follow.
+ Novelty: The idea itself is not exciting enough. It seems obvious that iteratively refining the initial prediction (anchors) can help to get better grounding results.


**Strength And Weaknesses:**

## Strengths
+ Extensive results are conducted on five benchmarks to validate the effectiveness of the LUNA model. The performance gains on the RefCOCO-family datasets are quite impressive.

## Weaknesses
+ The motivation of why need to introduce the anchor prior to the Transformer-based models is still unclear. From another perspective, the proposed continuous anchor-guided decoder (cf. Figure 2) can also be regarded as a prediction refinement network. In another world, the proposed decoder can also be used in other existing Transformer-based REC architectures to further boost their performance. It would be more interesting to see more results on other backbones.
+ For the results on Flickr30K Entities, it would be better to compare the LUNA to baselines with the same backbone (eg, ResNet-101).

**Summary Of The Paper:**

This paper focuses on the referring expression comprehension (REC) problem, ie, localizing the description of a natural language expression in an image. Specifically, they argue that existing Transformer-based methods don't use any location priors, which may result in inaccuracies in practice. To this end, they propose LUNA (LangUage as contiNuing Anchors) for Transformer-based REC models. Firstly, LUNA generates the first anchor box by attending to image regions under the guidance of the input expression. Then, in each layer, the anchor box is progressively updated into a new anchor. Extensive results on ReferIt, RefCOCO/+/g, and Flickr30K Entities have demonstrated the effectiveness of the proposed LUNA.

**Summary Of The Review:**

The paper argues that existing Transformer-based REC methods have overlooked the anchor priors. However, I think the motivation of this claim is still unclear. The whole proposed method is more like a post-processing step to further refine the grounding results. Although the proposed method achieves really good performance on these benchmarks, the idea itself is common and not exciting.

---

### Official Review · Reviewer_pGyA · 2022-10-23

**Confidence:** 3
**Correctness:** 3
**Technical Novelty And Significance:** 3
**Empirical Novelty And Significance:** Not applicable
**Recommendation:** 6

**Clarity, Quality, Novelty And Reproducibility:**

The paper is  well written, original and leverages marginally novel motivations and ideas to improve the tasks of one stage referring expression grounding.

**Strength And Weaknesses:**

Strengths - The method is simple and effective and relies on image and location only (without prior region / location information ) to learn grounding for referring expressions. This approach of using an initial proto-decoder to decode an estimate bounding box based on language and learning a continuous offset conditioned on prior knowledge is useful for together bounding box localization of referring expressions. They show several ablation studies and results on benchmark datasets to validate the effectiveness of the approach.

Weaknesses -
Is it possible to measure the performance of grounding just from the proto-decoder?This would tell how much gains are actually bought by the Anchor guided decoder ?

What if another prior / box location is used to guide the decoding process ? such as outputs from a pretrained system. In that case, the current method could also act as a corrective mechanism ?

What happens if the model predicts a completely different bounding box location in the proto-decoder stage?

Is there any benefit of splitting Q (Ix2C) to IxC and  IxC in Fig 3. (a) instead of learning projections ?


**Summary Of The Paper:**

In this paper, the authors propose to address referring expression grounding  using a single stage transformer based approach and leveraging anchor points (guided by language) as region priors. Their outputs are very well grounded with the attention maps learned during the decoding process and improves over state of the art for referring expression grounding on several benchmark datasets.

**Summary Of The Review:**

Please see the above section for details. The results and  method show improvements with interesting components that can be beneficial in other research works as well.

---

> ### Comment · Reviewer_pGyA · 2022-12-08
> **Feedback to authors response**
>
> I would like to thank the authors for their detailed response on the questions and additional experimental results to clarify the motivation and concerns raised.
>
> After another careful look at the responses and the concerns raised by other reviewers, these are my final thoughts. First, I still think that the motivation of the paper could be posed as a corrective mechanism (the iterative refinement strategy) rather than the proto-decoder of learning a position prior. This is just one of the many ways you could learn an initial anchor box.  I would agree with the reviewer uU4Z that the motivation hence is questionable and hence the novelty is marginal. Second, maybe there is another task or a weakly supervised setting where the importance of something like a proto-decoder can be shown effectively.
> This would make this paper demand a bit more work to make the contributions and motivation stronger. I would stick to my original rating of marginally above due to some important aspects such as improvements over the SOTA, ablation studies and careful analysis.

---

### Official Review · Reviewer_q7BF · 2022-10-25

**Confidence:** 4
**Correctness:** 3
**Technical Novelty And Significance:** 2
**Empirical Novelty And Significance:** 2
**Recommendation:** 5

**Clarity, Quality, Novelty And Reproducibility:**

This paper is well-written and has good clarity and reproducibility, yet the technical novelty should be emphasized to gain more support.


**Strength And Weaknesses:**

[Strengths]
+ This paper is well-written and easy to follow.
+ The ablation study and visualization are informative.

[Weaknesses]
- The proposed LUNA model integrates the BERT (Devlin et al.,2019) and ResNet-101 (He et al., 2016)/Swin-B (Liu et al., 2021), cross-attention (Vaswani et al., 2017), and DAB-DETR (Liu et al., 2022). Hence, the technical novelty is somewhat limited. It seems that the proto-decoder is the novel design of LUNA.
- The referring expression comprehension (RES) is one sort of cross-modal localization task. Therefore, after obtaining the initial localization via proto-decoder, employing the guided decoding mechanism proposed by DAB-DETR to refine the localization interactively is a straightforward solution. Is there any critical insight concerning the RES task to make LUNA different from DAB-DETR in the goal of localization block refinement?
- Table 1 misses the comparison with the recent methods (VILLA-large, MDETR, and [A]), which show better results than this work. Since the backbones (language and visual) and image resolutions are important factors of the RES task, it is better to indicate these factors for a fair comparison.

Related paper:

[A] Peng Wang, An Yang, Rui Men, Junyang Lin, Shuai Bai, Zhikang Li, Jianxin Ma, Chang Zhou, Jingren Zhou, Hongxia Yang: OFA: Unifying Architectures, Tasks, and Modalities Through a Simple Sequence-to-Sequence Learning Framework. ICML 2022: 23318-23340


**Summary Of The Paper:**

In order to address the referring expression comprehension task, this paper proposes the LangUage as contiNuing Anchors (LUNA) to employ the language-guided location priors to refine box prediction in a Transformer decoder progressively. Such an approach makes the decoder focus selectively on partial scenes and reduces noise in multi-modal contexts. Experiments show that LUNA achieves good referring expression comprehension performances.

**Summary Of The Review:**

The major concerns of this paper are its novelty and on-par performance, as described in [Weaknesses]. The proposed method integrates several recent models and hence achieves on-par performance. It is better to clarify its key insights and technical novelty to understand the contribution better. For a fair comparison, it is better to indicate the model settings of backbones and image resolutions for performance measurement.

---

### Decision · Program_Chairs · 2023-01-20

**Decision:**

Reject

**Justification For Why Not Higher Score:**

Reviewers reach a consensus on the above weaknesses regarding the novelty and motivation of this work. The author's response only addressed partial concerns while leaving the major concerns unresolved.

**Justification For Why Not Lower Score:**

N/A

**Metareview: Summary, Strengths And Weaknesses:**

This paper introduces a new method, Language as Continuing Anchors (LUNA), for the referring expression task, which employs language-guided location priors to refine box prediction progressively. The proposed method achieves new state of the art on multiple datasets.

Strengths:
- Strong experimental results for referring expression.
- The paper is well written.

Weaknesses:
- Reviewers have concerns about the technical contributions of this paper as the proposed method is an integration of existing methods.
- Lack of ablation study, especially on showing if the position prior is essential.
- The motivation of the method is unclear.